# Molecular epidemiology and long-term survival analysis of HIV-1/AIDS patients infected with CRF07_BC, CRF01_AE and subtype B in Taiwan

I-An Jen[1,2], Wei-You Li[3], Shang-Jung Wu[4], Patricia M. T. Chen[5], Wei-Lun Liu[6,7,8], Yung-Feng Yen[9], Cathy Shen Jang Fann[10], Yi-Ming Arthur Chen[3]*

**1** Institute of Brain Science, School of Medicine, National Yang Ming Chiao Tung University, Taipei, Taiwan, **2** Department and Institute of Public Health, School of Medicine, National Yang Ming Chiao Tung University, Taipei, Taiwan, **3** Graduate Institute of Biomedical and Pharmaceutical Science, College of Medicine, Fu Jen Catholic University, New Taipei City, Taiwan, **4** BioIT-Biostatistics Core Facility, Institute of Biomedical Sciences, Academia Sinica, Taipei, Taiwan, **5** University of California, Davis Medical Group, Sacramento, California, United States of America, **6** School of Medicine, College of Medicine, Fu Jen Catholic University, New Taipei City, Taiwan, **7** Department of Critical Care Medicine, Fu Jen Catholic University Hospital, Fu Jen Catholic University, New Taipei City, Taiwan, **8** Data Science Center, College of Medicine, Fu Jen Catholic University, New Taipei City, Taiwan, **9** Section of Infectious Diseases, Taipei City Hospital, Taipei, Taiwan, **10** Division of Infectious Disease and Immunology, Institute of Biomedical Sciences, Academia Sinica, Taipei, Taiwan

* 150110@mail.fju.edu.tw

## Abstract

Previously, we reported that HIV-1 patients infected with CRF07_BC had significantly lower viral loads than those infected with subtype B. Since HIV-1 viral load is associated with AIDS disease progression, the current study was to link multiple clinical and molecular databases, and compare clinical outcomes of HIV-1patients infected with CRF07_BC, CRF01_AE and subtype B in Taiwan. Molecular genotyping data of 2,982 HIV-1/AIDS patients were submitted to Taiwan CDC HIV-1/AIDS case management database. Then the database was linked to Taiwan National Health Insurance Research database and National Cause of Death database from 2000−2016. Subsequently, a subtype-based HIV/AIDS clinical database containing 1,605 patients including 858 (53.5%) subtype B, 690 (43.0%) CRF07_BC and 57 (3.5%) CRF01_AE patients was successfully established and the clinical outcomes and survival of these patients were analyzed. Analysis of transmission route showed this HIV-1/AIDS cohort consists of 761 (47.4%) men who have sex with men (MSM), 132 (8.2%) heterosexuals and 712 (44.4%) injection drug user (IDUs). Survival analysis showed subtype B patients had a significantly lower death rate (8.2%) than CRF07_BC and CRF01-AE patients (22.8% and 22.8%, respectively). The higher death rate for CRF07_BC versus subtype B patients could be largely influenced by transmission route (IDU: 95.7% vs. 3.9%; MSM: 85.8% vs. 2.0%), as well as lower ART uptake rates (69.9% vs. 96.3%). Indeed, subset analysis among IDU patients, CRF07_BC-infected patients had a better 16-year survival rate than patients infected

**Data availability statement:** Data cannot be shared publicly because of NHIRD and CDC regulations. Data are available from the NHIRD and CDC Institutional Data Access / Ethics Committee (contact via NHIRD ) for researchers who meet the criteria for access to confidential data. The data underlying the results presented in the study are available from NHIRD (https://nhird.nhri.edu.tw/en/).

**Funding:** This work was sponsored by the National Science and Technology Council, Taiwan, R.O.C. under Grant no. MOST 111-2327-B-016 -001 and by the Fu Jen Catholic University under Grant no. 913M332-01. The funders had no role in study design, data collection and analysis, decision to publish, or preparation of the manuscript.

**Competing interests:** The authors have declared that no competing interests exist.

with subtype B (74.3% vs. 45.7%, $p < 0.05$). Our findings suggest that the transmission route is one major factor influencing death rate, while ART treatment and HIV-1 subtypes may also play important roles. Our study is the largest long-term cohort study of patients infected with CRF07_BC and subtype B in the same geographic region.

## Introduction

Four decades after the first reported case of the acquired immunodeficiency syndrome (AIDS), people around the world continue to be baffled by how to control this disease and overcome the epidemic. One main reason is that human immunodeficiency virus type 1 (HIV-1), the causative organism of AIDS, presents with an extraordinary degree of genetic diversity. The genetic diversity of HIV-1 is mostly due to its high-mutation and recombination rates, large population size and rapid replication rate [1]. HIV-1 has been classified, based on phylogenetic clustering, into groups, subtypes and sub-subtypes [2]. There are four phylogenetically distinct groups (M, N, O, P) identified so far [3]. The predominant group M (main) has evolved into nine distinct subtypes (A–D, F–H, J, K), of which subtypes A and F have been further subdivided into sub-lineages (A1–A4, A6, and F1, F2) [4]. Furthermore, two or more subtypes of HIV-1 can combine to form a hybrid within a single infected individual, which leads to the emergence of additional novel variants such as circulating recombinant forms (CRFs) and unique recombinant forms (URFs) [5]. While a large number of novel HIV-1 CRFs and URFs were identified among injection drug users (IDUs), only few novel HIV-1 CRFs and URFs were found among patients infected through sexual contacts. According to an epidemiology report in 2019, almost half of all HIV-1 infections globally are of subtype C (47%), followed by subtype B (12%), subtype A (10%), and followed by CRF02_AG (8%) and CRF01_AE (5%) [6].

With the unique spatial quality of an island, Taiwan possesses a distinct HIV-1 epidemics. In Taiwan, the first HIV-1 case was reported in 1984. Since then, sexual transmission has been identified to be responsible for most HIV-1 infections in Taiwan. As reported by Chen YM et al., Taiwan's HIV-1 epidemic had initially spread primarily via sexual contact, with the subtype B and the circulating recombinant form CRF01_AE account for >95% of all infections [7]. However, since 2003, Taiwan has experienced a major outbreak of CRF07_BC, a circulating recombinant form of subtypes B and C, among IDUs [8,9]. Intravenous drug use has since emerged as a new route of HIV-1 transmission and contributed to a substantial increase in Taiwanese HIV-1 cases [9]. Molecular epidemiological studies also demonstrated more than 95% of IDUs with newly diagnosed HIV-1 in 2004 and 2005 were infected with CRF07_BC [9]. Previously, we reported that HIV-1/AIDS patients infected with CRF07_BC had significantly lower viral loads than those infected with subtype B [10]. However, the number of patients in the previous study was relatively small (21 CRF07_BC infected patients and 59 subtype B infected patients), and with a relatively short follow up period (3 years). The aim of this study is to link our molecular

genotyping database with multiple clinical databases including treatment and death record databases to compare clinical outcomes and survival among HIV-1 patients with different transmission routes (men who have sex with men [MSM], IDU versus heterosexuals) infected with different subtypes including CRF01_AE, CRF07_BC and subtype B. Since the extent of viremia, measured as HIV-1 RNA viral load, has been postulated to be the best available surrogate marker of HIV-1 disease progression [11], we hypothesized that patients infected with CRF07_BC might have better survival than patients infected with subtype B if they share similar transmission risk factors. In the current study, we analyzed 16-year survival of Taiwanese patients infected with CRF07_BC, CRF01_AE and subtype B with similar risk factors.

## Materials and methods

**Linkage analysis of HIV-1 subtype database with Taiwan** CDC HIV/AIDS case management database, cause of death record database and National Health Insurance Research Database (**NHIRD).**

As depicted in the flowchart of Fig 1, the HIV-1 subtype information was collected from a collaborative HIV-1 archive of the Chen's lab (n = 2,982) [7–10,12–16]. The database of HIV-1/AIDS patients with subtypes were uploaded to a HIV/AIDS case management database of Taiwan CDC manually based on the ID information of individuals. After that, these results were combined into each individual's profile as a new column of data. Furthermore, in order to execute a comprehensive analysis, the data of HIV-1 subtype, viral load and CD4, the medical record were output from Taiwan CDC HIV/AIDS case management database. The cause of death record was output from the Cause of death data of Department of Statistics, Taiwan Ministry of Health and Welfare (MOHW). The medication and comorbidity records were output from Taiwan National Health Insurance Research Database (NHIRD). Data collected from these three different sources were then integrated into a single database and matched via individual's ID information. The Taiwan NHIRD was used to screen for all people living with HIV/AIDS (PLWHA) from January 1, 2000 to December 31, 2016 by using ICD-9-CM and ICD-10-CM codes. A total of 32,390 cases were then matched to the confirmed HIV-1 positive cases in the HIV/AIDS registry of Taiwan CDC. Three main subtypes were identified in 2,982 patients. Cases with incomplete demographic data were excluded, yielding a total of 1,605 adults HIV-infected cases with subtype information. Altogether, there were 1,605 subtype data matched to the individual's profile in Taiwan NHIRD and cause of death data successfully. Late HIV diagnosis

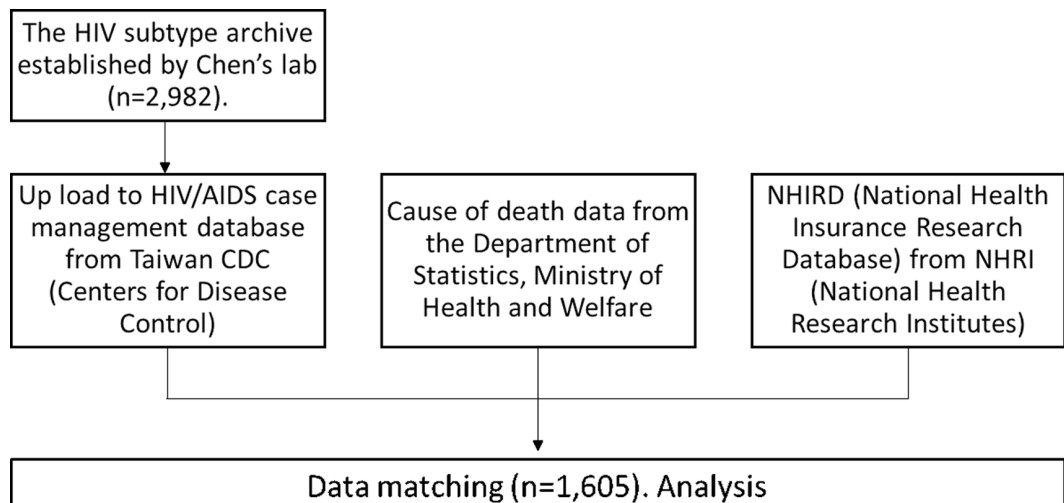

**Fig 1. The flowchart of HIV-1 subtypes information collection and linking with HIV/AIDS case management database from Taiwan CDC, Cause of death data from the Department of Statistics, and NHIRD.**

is defined as having AIDS at the time of initial HIV disease diagnosis or being diagnosed with AIDS within 12 months of initial diagnosis [17]. We started to access the dataset at 21/10/2016 and continued the analysis till 2022.

## Determination of HIV-1 genotypes

The genotype of HIV-1 infection was determined by using different methods including nested multiplex PCR and nucleic acid sequencing combined with phylogenetic tree analysis [7–10,12–16]. In brief, blood samples from participants were collected and DNA was extracted from peripheral blood mononuclear cells. The HIV-1 gag gene regions were amplified using PCR. Nested multiplex PCR was performed using rTaq DNA Polymerase (TOYOBO CO., LTD, Japan) with primers designed based on subtype-specific segments in the gag region [18]. Different HIV-1 subtypes produce product segments of varying sizes which can be used to determine the specific subtype. Once multiplex PCR showed two or more HIV-1 subtypes suggesting dual infection, serial PCR using a single pair of subtype-specific primers was used to confirm dual infection. To confirm the results of multiplex PCR, representative samples were selected for nucleic acid sequencing combined with phylogenetic tree analysis [14–16].

## Statistical analyses

Statistical analysis was performed with SAS version 9.4, and statistical significance was set at $p < 0.05$. The registry automatically de-identified all the cases used in the statistical analysis, and the actual number of cases would not show in any of the statistical analysis results if it was less than 3 so that individual cases would not be identified or inferred. Therefore, use of the Taiwan NHIRD does not require any signed consent. Kaplan–Meier curves were used to estimate cumulative survival in different subgroups of HIV infected patients classified by subtypes, routes of transmission and with or without antiretroviral therapy (ART).

## Ethical clearance

The Institutional Review Board of Taipei City Hospital Research Ethics Committee and NHIRD approved the study and waived the need for informed consent, all research was performed in accordance with relevant guidelines.

## Results

### Demographic characteristics of three main HIV-1 subtypes in Taiwan

All HIV-infected individuals reported to Taiwan CDC from 2000 to 2016 were initially enrolled in the current study. After excluding individuals younger than 15 years and those with incomplete data, the remaining 32,390 people living with HIV/AIDS (PLWHA) constituted the study population (Table 1). Three main HIV-1 subtypes, namely, subtype B, CRF07_BC and CRF01_AE, were identified. Two HIV-1-infected patients who were dually infected with CRF01_AE and subtype B have been excluded from the survival analysis to avoid misclassification and confusion. The results showed while subtype B (858 patients, 53.5%) and CRF07_BC (690 patients, 43.0%) constituted the bulk, CRF01_AE (57 patients, 3.5%) was substantially less prevalent among the 1,605 patients with available HIV-1 subtype information. The mean age of all patients was 32.5 ± 10.1 years. Most of the genotyped HIV-1 patients were in the age groups of 21−30 and 31−40 (50.8% and 32.1%, respectively), and of male gender (92.4%). In terms of socioeconomic status, most genotyped patients were in the low-income group (66.2%) and from urban location (73.5%). As far as the route of HIV-1 transmission in patients who were genotyped, MSM, heterosexuals and IDUs accounted for 47.4%, 8.2% and 44.4%, respectively (Table 1).

By comparing demographics between these subtypes, subtype B patients were relatively younger (29.7 ± 7.9 years), had a lesser prevalence in low-income population (49.0%), and had a greater prevalence in both urban location (85.4%) and transmission through homosexual contact (85.8%). In comparison, CRF07_BC and CRF01_AE patients were slightly older (32.2 ± 7.6 and 34.8 ± 9.1 years, respectively), and had a greater prevalence in low-income population (87.1% and

**Table 1. Demographics of three HIV-1 subtype cohorts and not genotyped group in Taiwan.**

| Cohorts Characteristics | Subtype B (N = 858) | CRF07_BC (N = 690) | CRF01_AE (N = 57) | Not genotyped (N = 30785) | All (N = 32390) |
|---|---|---|---|---|---|
| Age, mean (year) | 29.7 ± 7.9 | 32.2 ± 7.6 | 34.8 ± 9.1 | 32.5 ± 10.2 | 32.5 ± 10.1 |
| Age group | | | | | |
| ≦ 20 | 55 (6.4%) | 14 (2.0%) | 3 (5.3%) | 1687 (5.5%) | 1759 (5.4%) |
| 21-30 | 475 (55.4%) | 320 (46.4%) | 21 (36.8%) | 13958 (45.3%) | 14774 (45.6%) |
| 31-40 | 252 (29.4%) | 243 (35.2%) | 20 (35.1%) | 9376 (30.5%) | 9891 (30.5%) |
| 41-50 | 59 (6.9%) | 95 (13.8%) | 9 (15.8%) | 3916 (12.7%) | 4079 (12.6%) |
| ≧ 51 | 17 (2.0) | 18 (2.6%) | 4 (7.0%) | 1848 (6.0%) | 1887 (5.8%) |
| Gender | | | | | |
| Male | 845 (98.5%) | 592 (85.8%) | 46 (80.7%) | 28933 (94.0%) | 30416 (93.9%) |
| Female | 13 (1.5%) | 97 (14.1%) | 11 (19.3%) | 1776 (5.8%) | 1897 (5.9%) |
| Unknown | 0 (0.0%) | 1 (0.1%) | 0 (0.0%) | 76 (0.3%) | 77 (0.2%) |
| Income level | | | | | |
| Low | 420 (49.0%) | 601 (87.1%) | 42 (73.7%) | 20704 (67.3%) | 21767 (67.2%) |
| Intermediate | 322 (37.5%) | 72 (10.4%) | 9 (15.8%) | 7194 (23.4%) | 7597 (23.5%) |
| High | 116 (13.5%) | 10 (1.5%) | 6 (10.5%) | 2740 (8.9%) | 2872 (8.9%) |
| Unknown | 0 (0.0%) | 7 (1.0%) | 0 (0.0%) | 147 (0.5%) | 154 (0.5%) |
| Geographic location | | | | | |
| Rural | 125 (14.6%) | 282 (40.9%) | 18 (31.6%) | 9870 (32.1%) | 10295 (31.8%) |
| Urban | 733 (85.4%) | 408 (59.1%) | 39 (68.4%) | 20915 (68.0%) | 22095 (68.2%) |
| Transmission route | | | | | |
| Homosexual contact | 736 (85.8%) | 14 (2.0%) | 11 (19.3%) | 18880 (61.3%) | 19641 (60.6%) |
| Heterosexual contact | 89 (10.4%) | 16 (2.3%) | 27 (47.4%) | 5275 (17.3%) | 5407 (16.7%) |
| IDU* | 33 (3.9%) | 660 (95.7%) | 19 (33.3%) | 6236 (20.3%) | 6948 (21.5%) |
| Unknown | 0 (0.0%) | 0 (0.0%) | 0 (0.0%) | 394 (1.3%) | 394 (1.2%) |

*IDU = Injection drug user.

73.7%, respectively), and a lesser prevalence in both urban location (59.1% and 68.4%, respectively) and transmission through homosexual contact (2.0% and 19.3%, respectively). In particular, the majority of CRF07_BC patients (95.7%) had IDU as the transmission route.

**Antiretroviral therapy (ART) and outcomes among three main HIV-1 subtypes**

As shown in Table 2, 80.2% (21,019/26,207) of the HIV-1 patients with complete medical records in Taiwan CDC database received ART by the end of 2016. Among them, only 69.9% of CRF07_BC patients received ART, which is significantly lower than 96.3% for subtype B patients and 94% for CRF01_AE patients (p < 0.001). In terms of the mean CD4 count at their first visit to the AIDS clinics, CRF07_BC patients had significantly lower number of CD4 count than the other two cohorts (p < 0.001). It is important to note that the mean CD4 count for all three cohorts at their last clinical visits increased, especially for subtype B patients. This phenomenon was also reflected in the CD4 count stratification, e.g., for subtype B patients, 8.3% of them had CD4 count less than 200 during their first clinic visit, which decreased to 5.1% in their last visit. Reviewing the percentage of patients with either an unchanged or an increased in their CD4 count from their first clinic visit to their last clinic visit, HIV-1 patients with CRF07-BC have the least number of patients with either an increased or unchanged CD4 count compared to the other two groups of patients.

Table 3 showed comparison of ART treatment outcomes between 778 subtype B and 414 CRF07_BC patients who received ART during this period of time. The results showed the mean number of the first CD4 count of subtype B patients

**Table 2. Comparison of the rates of antiretroviral therapy (ART) and CD4 counts of the first and last tests among three HIV-1 subtype cohorts and not being genotyped patients in Taiwan.**

| Characteristic | Subtype(+) | | | Not genotyped | TOTAL | P value |
|---|---|---|---|---|---|---|
| | Subtype B | (CRF)07_BC | (CRF)01_AE | | | |
| Variable | N = 808 | N = 592 | N = 50 | N = 24757 | N = 26207 | |
| ART No. (%) | | | | | | |
| No | 30 (3.7) | 178 (30.1) | 3 (6.0) | 4977 (20.1) | 5188 (19.8) | <0.001** |
| Yes | 778 (96.3) | 414 (69.9) | 47 (94.0) | 19780 (79.9) | 21019(80.2) | |
| CD4 counts of the first test, mean (±standard deviation/mm3) | 571.41 (±307.95) | 418.14 (±200.20) | 471.31 (±283.25) | 416.79 (±785.94) | 421.7 (±767.0) | <0.001** |
| Groups of CD4 counts of the first test, N (%) | | | | | | |
| <200 | 67 (8.29) | 61 (10.30) | 8 (16.00) | 4597 (18.57) | 4733(18.06) | <0.001** |
| 200-350 | 123 (15.22) | 194 (32.77) | 12 (24.00) | 6320 (25.53) | 6649(25.37) | |
| 351-499 | 170 (21.04) | 170 (28.72) | 11 (22.00) | 6247 (25.23) | 6598(25.18) | |
| >=500 | 447 (55.32) | 164 (27.70) | 19 (38.00) | 7571 (30.58) | 8201(31.29) | |
| NA* | 1 (0.12) | 3 (0.51) | 0 (0.00) | 22 (0.09) | 26 (0.10) | |
| CD4 counts of the last test, mean (±standard deviation/mm3) | 667.35 (±322.60) | 462.29 (±245.92) | 544.58 (±329.16) | 558.77 (±256.21) | 560.0 (±249.3) | 0.5 |
| Groups of CD4 counts of the last test, N (%) | | | | | | |
| <200 | 41 (5.07) | 61 (10.30) | 6 (12.00) | 1824 (7.37) | 1932 (7.37) | <0.001** |
| 200-350 | 72 (8.91) | 138 (23.31) | 10 (20.00) | 3942 (15.92) | 4162 (15.88) | |
| 351-499 | 133 (16.46) | 178 (30.07) | 8 (16.00) | 5691 (22.99) | 6010 (22.93) | |
| >=500 | 553 (68.44) | 192 (32.43) | 24 (48.00) | 12505 (50.51) | 13274 (50.65) | |
| NA | 9 (1.11) | 23 (3.89) | 2 (4.00) | 795 (3.21) | 829 (3.16) | |

NA: not available

was significantly higher than that of CRF07_BC patients (p<0.001). Both treatment cohorts' mean CD4 count increased at their last AIDS Clinics visit. By stratifying patient's CD4 count into 4 levels, it was clearly demonstrated that subtype B patients had better treatment responses than the CRF07_BC patients. For example, the number of long-term nonprogressors (LTNPs, CD4 count >500WBC/mm$^3$) for subtype B patients increased 13.8% (from 55.5% to 69.3%). In contrast, the number of LTNPs for CRF07_BC patients only increased 9.4% (from 22.2% to 31.6%). It is also worth noting that patients in both cohorts responded to ART and had less patients with CD4 count less than 200 WBC/mm$^3$ at their last clinical visit. The number of AIDS patients for both cohorts also decreased: subtype B patients decreased 3.6% (from 8.6% to 5.0%) and CRF07_BC patients decreased 1.5% (from 13.3% to 11.8%).

### Death rates of HIV-1 patients among three main HIV-1 subtypes and different demographic groups in Taiwan

The overall death rate for all HIV-1 patients in our study was 14.0% (Table 4). Among three main HIV-1 subtypes, subtype B patients had significantly lower death rate of 8.2%, comparing to 22.8% for both CRF07_BC and CRF01_AE patients. Patients with older age, female gender and low-income level were found to have higher death rates. As shown in Table 4, there was a steady increase in death rate from 7.1% for age group ≤20 to 7.7%, 15.1%, 24.9% and 40.9% for ages groups 21−30, 31−40, 41−50 and ≥51, respectively. In our study, female patients exhibited a higher death rate than male patients (20.2% vs. 13.6%). In addition, low-income level was associated with a higher death rate than high-income level (15.9% vs. 8.7%). However, there was only little difference in death rate between patients residing in rural and urban locations (15.4% vs. 13.4%). Among different routes of HIV-1 transmission, homosexual contact was associated with significantly lower death rate of 5.5%, comparing to 27.6% and 24.5% for IDU and heterosexual contact, respectively (Table 4).

**Table 3. Comparison of the mean CD4 counts of the first and last tests and the treatment outcome between HIV-1 subtype B and CRF07_BC patients who have received ART in Taiwan.**

| Subtype Cohorts | Subtype B | (CRF)07_BC | *P* value |
|---|---|---|---|
| CD4 counts (No./mm3) | (n = 778) | (n = 414) | |
| **Mean CD4 counts of the first test** (±standard deviation) | 573.8 (±312.07) | 388.51 (±194.73) | <0.001** |
| **Groups of CD4 counts of the first test, N (%)** | | | |
| <200 | 67 (8.61) | 55 (13.29) | <0.001** |
| 200-350 | 118 (15.17) | 153 (36.96) | |
| 351-499 | 160 (20.57) | 112 (27.05) | |
| >=500 | 432 (55.53) | 92 (22.22) | |
| NA* | 1 (0.13) | 2 (0.48) | |
| **Mean CD4 counts of the last test** (±standard deviation) | 673.64 (±323.6) | 456.48 (±252.34) | <0.001** |
| **Groups of CD4 counts of the last test, N (%)** | | | |
| <200 | 39 (5.01) | 49 (11.84) | <0.001** |
| 200-350 | 66 (8.48) | 98 (23.67) | |
| 351-499 | 127 (16.32) | 127 (30.68) | |
| >=500 | 539 (69.28) | 131 (31.64) | |
| NA* | 7 (0.90) | 9 (2.17) | |
| **CD4 counts comparison between the first and last tests N (%)** | | | |
| Unchanged | 467 (60.03) | 186 (44.93) | <0.001** |
| high increase | 49 (6.30) | 30 (7.25) | |
| low increase | 176 (22.62) | 110 (26.57) | |
| decrease | 78 (10.03) | 77 (18.60) | |
| NA | 8 (1.03) | 11 (2.66) | |

NA: not available.

## Factors associated with clinical outcomes of HIV-1 patients in Taiwan

Table 5 showed the clinical outcomes of HIV-1 patients in Taiwan, grouped by HIV-1 subtype. In terms of late HIV-1 diagnosis, patients with unknown subtype had a substantially higher rate of 28.5%. In comparison, patients with known subtype information were less likely to have late HIV diagnosis (12.0%, 3.5% and 7.0% for subtype B, CRF07_BC and CRF01_AE, respectively; Table 5). In terms of death rate, while higher rates were observed in CRF07_BC patients (29.2% and 22.5% for with and without late HIV diagnosis, respectively) and CRF01_AE patients without late HIV diagnosis (24.5%), subtype B patients had relatively lower death rates (8.7% and 8.1% for with and without late HIV diagnosis, respectively). The death rates for patients with unknown HIV-1 subtype were 17.5% and 12.7% for with and without late HIV diagnosis, respectively (Table 6).

Kaplan-Meier curves of 1,605 HIV-1 patients in Taiwan with three main subtypes, grouped by ART usage status, were generated for survival analysis (Fig 2). Among the three subtypes, subtype B patients had the most favorable 16-year survival of 90.0%, comparing to 73.3% for patients with CRF07_BC and CRF01_AE (Fig 2A). Regarding ART usage, patients received ART showed generally better survivals than patients did not receive ART irrespective of different subtypes (Fig 2B and 2C). The 16-year survivals for ART-treated patients were 92.4%, 79.5% and 82.8% for subtype B, CRF07_BC and CRF_AE. In comparison, the 16-year survivals for no ART-treated patients were 77.0%, 70.7%, 19.0% for subtype B, CRF07_BC and CRF01_AE.

**Table 4. Death rates of three main HIV-1 subtypes and among different demographic groups in Taiwan.**

| Characteristics | Death rate (Death No./ Patient No.) | Multivariate analysis HR (95%CI) | p value |
|---|---|---|---|
| All patients | 14.0% (4546/ 32390) | | |
| Subtype | | | |
| subtype B | 8.2% (70/ 858) | | |
| CRF07_BC | 22.8% (157/ 690) | 1.27 (0.95-1.69) | --- |
| CRF01_AE | 22.8% (13/ 57) | 1.25 (0.69-2.26) | --- |
| Unknown subtype | 14.0% (4306/ 30785) | 1.65 (1.30-2.09) | < 0.01 |
| Age group (years) | | | |
| ≦ 20 | 7.1% (124/ 1759) | | |
| 21-30 | 7.7% (1140/ 14774) | 1.67 (1.36-2.05) | < 0.01 |
| 31-40 | 15.1% (1495/ 9891) | 2.68 (2.19-3.29) | < 0.01 |
| 41-50 | 24.9% (1015/ 4079) | 4.31 (3.50-5.30) | < 0.01 |
| ≧ 51 | 40.9% (772/ 1887) | 7.47 (6.05-9.22) | < 0.01 |
| Gender | | | |
| Male | 13.6% (4148/ 30416) | | |
| Female | 20.2% (383/ 1897) | 0.71 (0.64-0.79) | < 0.01 |
| Unknown | 19.5% (15/ 77) | 1.59 (0.67-3.75) | --- |
| Income level | | | |
| Low | 15.9% (3470/ 21767) | | |
| Intermediate | 10.6% (803/ 7597) | 0.83 (0.76-0.89) | < 0.01 |
| High | 8.7% (250/ 2872) | 0.63 (0.55-0.72) | < 0.01 |
| Unknown | 14.9% (23/ 154) | 0.80 (0.41-1.58) | --- |
| Geographic location | | | |
| Rural | 15.4% (1581/ 10295) | | |
| Urban | 13.4% (2965/ 22095) | 0.96 (0.90-1.02) | --- |
| Transmission route | | | |
| Homosexual contact | 5.5% (1085/ 19641) | | |
| Heterosexual contact | 24.5% (1322/ 5407) | 2.33 (2.13-2.54) | < 0.01 |
| IDU* | 27.6% (1914/ 6948) | 2.76 (2.54-3.00) | < 0.01 |
| Unknown | 57.1% (225/ 394) | 15.7 (13.4-18.3) | < 0.01 |

*IDU = Injection drug user.

**Table 5. The rate of late HIV diagnosis of three main HIV-1 subtypes in Taiwan.**

| Subtype | Rate of late HIV diagnosis (Diagnosis No. / Patient No.) |
|---|---|
| subtype B | 12.0% (103 / 858) |
| CRF07_BC | 3.5% (24 / 690) |
| CRF01_AE | 7.0% (4 / 57) x |
| Unknown subtype | 28.5% (8774 / 30785) |

Fig 2D–2F showed the Kaplan-Meier survival curves of HIV-1 patients in Taiwan with three main subtypes, grouped by different transmission routes. Although there was no discernible difference in survivals among three subtypes of patients with HIV-1 transmitted through homosexual and heterosexual contacts, IDU patients showed significant worse 16-year

**Table 6. The rate of death in relation to late HIV diagnosis of three main HIV-1 subtypes in Taiwan.**

| Subtype | Death rate (Death No. / Patient No.) | |
| --- | --- | --- |
| | **With late HIV diagnosis** | **Without late HIV diagnosis** |
| subtype B | 8.7% (9 / 103) | 8.1% (61 / 755) |
| CRF07_BC | 29.2% (7 / 24) | 22.5% (150 / 666) |
| CRF01_AE | 0.0% (0 / 4) | 24.5% (13 / 53) |
| Unknown subtype | 17.5% (1562 / 8905) | 12.7% (2984 /23485) |

survivals. It is important to note that the 16-year survivals for IDU-transmitted patients were 45.7%, 74.3% and 57.0% for subtype B, CRF07_BC and CRF01_AE and the differences were statistically significant (Fig 2F). The hazard ratio for the 16-year mortality rate is 3.84 (95% Confidence Interval: 2.68–5.49) for CRF07_BC and 5.43 (95% Confidence Interval: 2.78–10.64) for CRF01_AE compared with subtype B (Table 7).

## Discussion

In this study, we found most patients with HIV-1 diagnosis in Taiwan from 2000 to 2016 were of younger ages between 21 and 40 years old (21−30 years old, 45.6%; 31−40 years old, 30.5%), only few patients were over the age of 50 (5.8%). The mean age for all patients were 32.5 years old. This is true in all three known HIV-1 subtypes and unknown subtype patients. Among them, subtype B and CRF01_AE have the youngest and oldest mean ages of 29.7 and 34.8 years old, respectively. This finding is consistent with the 2020 US CDC data, which reported young people aged 13–24 were especially vulnerable to HIV-1 infection and accounted for 20% (6,135) of all new HIV diagnoses in US [19]. In addition, we found that HIV-1 patients in Taiwan, regardless of the subtypes, predominantly were male (93.9%), and mostly were associated with low income (67.2%) and lived in the urban area (68.2%). Difficulties in accessing HIV care in the rural areas may explain why majority of HIV-1 positive patients lived in urban areas.

Interestingly, the transmission route for HIV-1 in Taiwan was different among its three major subtypes. While subtype B patients were mostly transmitted through homosexual contact (85.8%), CRF07_BC patients were mostly transmitted through intravenous drug use (IDU, 95.6%), and CRF01_AE patients were mostly transmitted through heterosexual contact (47.4%). These findings were consistent with our previous study of HIV-1 patients in Taiwan, in which MSM (men having sex with men) patients were mostly infected with subtype B; heterosexual patients were mostly infected with CRF_01 AE, and most of the IDU patients were infected with CRF07_BC [9,14]. As for all three subtypes and unknown subtype HIV-1 patients, homosexual contact (60.6%) was the most common route of HIV-1 transmission in Taiwan.

In our study cohort, 80.2% of HIV-1 patients in Taiwan took ART (antiretroviral therapy). Among all subtypes, most subtype B patients (96.3%) took ART, followed by CRF01_AE (94%), and, lastly, CRF07_BC (69.9%). In this regard, UNAIDS reported 28.7 million people with HIV-1 (75%) had access to antiretroviral therapy (ART) globally as of the end of 2021 [20]. Therefore, a higher percentage of HIV-1 patients in Taiwan took ART compared to globally. This favorable situation probably is a direct result from the Taiwanese government adopting a policy since April 1997 to provide all HIV-1 infected citizens with free access to HAART (Highly Active Antiretroviral Therapy) through the National Health Insurance program. In order to maximize the benefits of ART, Taiwanese government also created special clinics where antiretroviral agents could be prescribed and their use monitored by qualified physicians [21]. Having this free and dedicated access to ART certainly can enhance higher percentage of HIV-1 patients to seek ART treatment compared to globally.

Our data also showed older patients with HIV-1 were association with higher mortality rates in Taiwan, particularly for patients older than 50 years old (Table 4). In 2016, Asher et al. studied clinical, immunological and virologic status in patients diagnosed with HIV above the age of 50. They found that older patients presented with significantly lower CD4 cell counts and higher viral load compared with younger patients. Furthermore, they found patients older than 50 years old had significantly higher mortality rate than younger patients (21% vs. 3.5%, p<0.001) [22]. Our results are consistent with

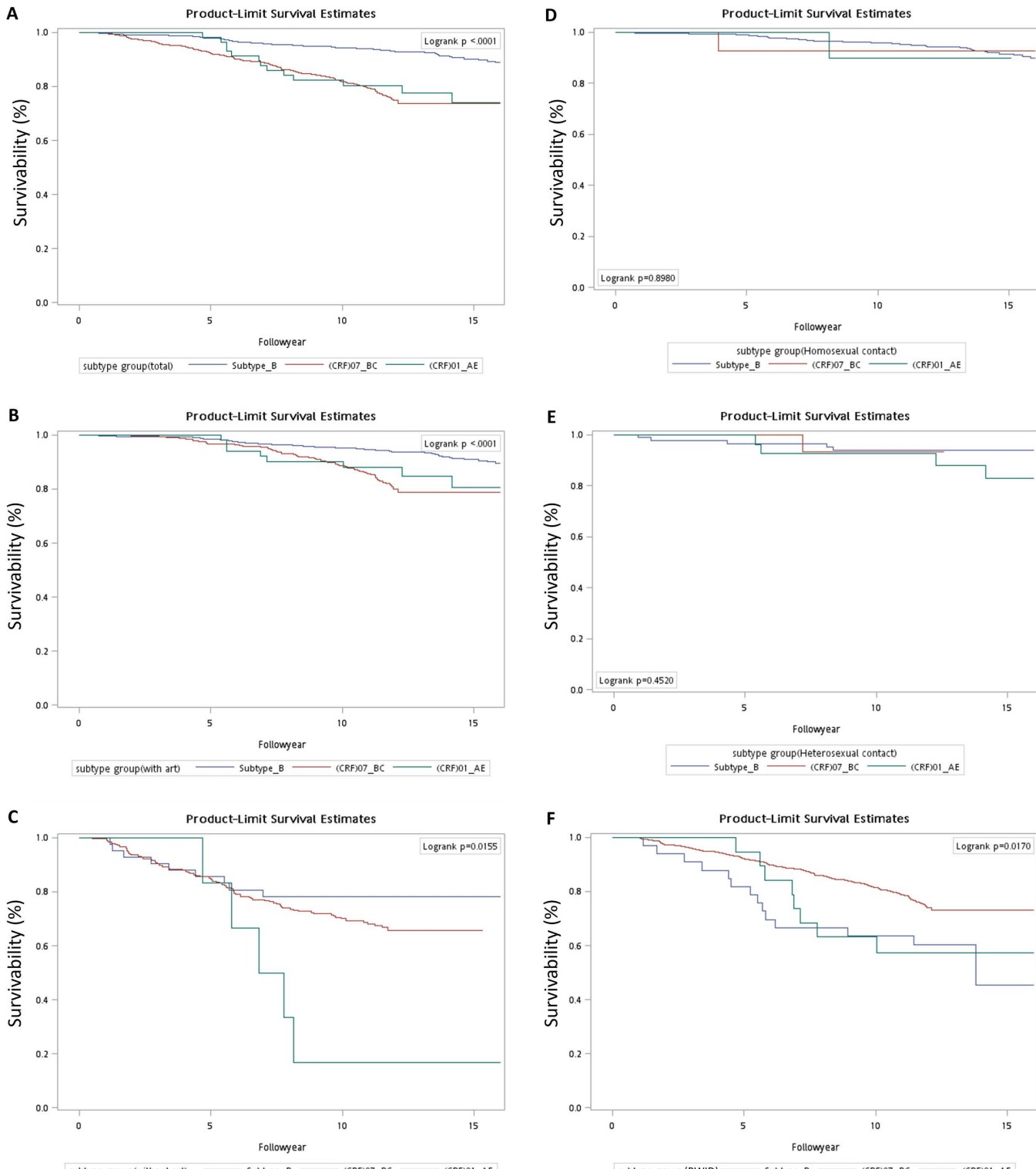

**Fig 2. Overall survival of Taiwanese patients infected with three main HIV-1 Subtypes grouped by antiretroviral therapy (ART) usage status: (A) all patients, (B) patients received ART, (C) patients didn't receive ART.** Overall survival of Taiwanese patients infected with three main HIV-1 subtypes grouped by route of transmission: (D) homosexual contact, (E) heterosexual contact, and (F) injection drug user (IDU).

**Table 7. The survival analysis of three main HIV-1 subtypes in Taiwan.**

|  | 16-year survival rate | Hazard ratio | CI (%) |
|---|---|---|---|
| subtype B | 91.8% (788/858) | -- | -- |
| CRF07_BC | 77.2% (533/690) | 3.84 | (2.68-5.49) |
| CRF01_AE | 77.2% (44/57) | 5.43 | (2.78-10.64) |

their findings. In our study, patients older than 50 years old showed much higher mortality rate than patients in the age group of 21–30 years old (40.9% vs. 7.7%, p<0.001).

Interestingly, if we don't consider the risk factors of transmission routes first, we found that overall, CRF07_BC has a significantly higher death rate (22.8% vs. 8.2%) compared to subtype B patients. The 16-year survivals with and without ART were also lower for CRF07_BC compared to subtype B (with ART: 79.5% versus 92.4%; without ART: 70.7% versus 77.0%). It's intriguing why CRF07_BC patients with likely lower viral load but with higher death rate. One plausible explanation is that since patients with CRF07_BC were mostly IDUs in Taiwan, other co-morbidities associated with IDUs such as unintentional overdose, hepatitis C, invasive bacteria, or fungal infection and acute or chronic pulmonary complications may contribute to their higher death rate. In contrast, subtype B patients were mostly homosexual men who attended our outreach programs and received counseling when they were first diagnosed HIV-1 disease. The fact that majority of the subtype B HIV-1 patients participated in the free ART program (Table 2). However, the interval between initiation of ART and the onset of HIV-1 infection is difficult to estimate in most of the scenario except in the blood transfusion infection as we've reported before [23]. Therefore, even we've realized the relative timing of ART initiation in disease progression is important, we still have to tolerate with this variation factor to execute our analysis.

Previously published works have shown no difference in either the rate of immunological progression prior to ART or the initial virologic response to ART between subtype B and all non-B subtypes combined [24,25]. Notably, in these studies, the non-B subtypes of HIV-1 patients included only Subtype A, C, D, and CRF02AG, and did not include CRF07_BC or CRF_01 AE subtypes. It remains plausible that CRF07_BC or CRF_01 AE subtypes can harbor an inherited variation that causes a lower degree of immunological response from the host, hence a lesser virologic response to ART compared to subtype B. In this regard, our previous study showed that 21 patients infected with CRF07_BC had similar CD4 count, but significantly lower viral load than 59 patients infected with subtype B [10]. Further *in vitro* experiments utilizing mutated HIV-1 recombinant viruses in the same study provided strong mechanistic evidence that a 7 amino-acid deletion of p6$^{gag}$ in the HIV-1 genome of CRF07_BC, which could cause a disruption in virus assembly, budding, maturation and release of HIV-1 virus, was the main culprit for the lower viral load in patients infected with CRF07_BC [10]. Known as a major prognostic factor, lower viral load in patients infected with CRF07_BC, theoretically, should be associated with slower disease progression rate, as well as lower death rate than patients infected with subtype B.

In this study, we found that untreated CRF07_BC patients had lower mean viral loads (31,298±3706) than patients infected with subtype B (56,658±11,275) during their first clinical visit (S1 Table). Moreover, among the IDU-transmitted patients who had been genotyped, CRF07_BC patients had better survival rate than patients infected with subtype B or CRF01_AE and the differences are statistically significant (Fig 2F). These findings support our hypothesis that under the premise of same risk factor, a lower viral load in CRF07_BC patients can contribute to their slower disease progression. This finding is consistent with the observation from researchers in mainland China, showing CRF07_BC is one of the four major subtypes with the best outcome and slowest disease progression [26,27]. As we know, the major transmission risk factor of CRF07_BC in northern mainland China is MSM, and in southwestern being heterosexual contact and MSM [28,29]. Given the complexity of socioeconomic status, legal status, and comorbidities in the IDU group comparing to the MSM and heterosexual contact groups, the viral genetic effect on disease progression could be easier to observe than in

Taiwan. Our study is the first to provide clinical evidence suggesting genetic alterations in CRF07_BC subtype may play a role in determining HIV-1/AIDS disease progression.

Late HIV diagnosis generally represents missed opportunities to treat and prevent HIV-1 infection. In our study population, 27.5% of Taiwanese HIV-1 patients were identified to have late HIV diagnosis. Patients with late HIV diagnosis had a higher death rate than patients without late HIV diagnosis (17.5% versus 12.7%). Among patients with known HIV-1 subtypes, patients with subtype B had the highest rate of late HIV diagnosis. Given subtype B patients in our study were mostly homosexuals, these patients probably did not seek medical attention until late in the disease process due to fear of stigma and discrimination associated with homosexuality. Conversely, HIV-1 patients with subtype CRF_07BC had the lowest rate of late HIV diagnosis. This is likely because most of our patients with CRF_07BC subtype were IDUs and frequently incarcerated, which led to earlier detections of HIV-1 disease when they underwent mandatory HIV testing during imprisoning.

Our study has some limitations. We had only few patients in the CRF01AE subtype. This certainly presents a challenge in the analysis. Though highly unlikely, possibility exists that other dominant subtype(s) of HIV-1 may reside within the large number of patients with unknown subtype(s). We do not have the complete laboratory data on CD4 counts or viral loads of the HIV-1 patients due to the unavailability of these data from Taiwan CDC. Although we do not have the subtype information on a rather large number of HIV-1 patients in Taiwan, we believe the 690 CRF07_BC patients in this study represent the currently largest cohort with a 16-year long-term follow up.

In conclusion, our study is the first nationwide population-based molecular cohort study for both demographics and clinical outcomes of Taiwanese HIV-1 patients from 2000 to 2016. Our study provides detailed information on the three main HIV-1 subtypes, namely, subtype B, CRF07_BC and CRF01_AE in Taiwan. The results showed that most Taiwanese HIV-1 patients were in a relatively younger population, and there was an age dependent association between older patients and higher death rates in these patients. Among the three main HIV-1 subtypes, while subtype B patients had a significantly lower death rate (8.2%) than CRF07_BC and CRF01-AE patients (22.8% and 22.8%). The higher death rate for CRF07_BC versus subtype B seems to be influenced by the mode of transmission (IDU vs. MSM), as well as lower ART uptake rates (69.9% vs. 96.3%). Among the HIV-1 patients contracted through IDU, CRF07_BC patients had better 16-year survival than patients infected with subtype B (74.3% vs. 45.7%, $p < 0.05$). These findings suggest that the transmission route is one major factor influencing death rate. Our findings highlight the importance of molecular genotyping for clinical counseling and patient managements in Taiwan and other geographic regions where multiple HIV-1 genotypes exist.

## Supporting information

**S1 Table. Comparison of the mean CD4 counts of the first and last tests between HIV-1 subtype B and CRF07_BC patients who did not receive ART in Taiwan.**
(DOCX)

**S1 Dataset. Minimal dataset.**
(ZIP)

## Author contributions

**Conceptualization:** I-An Jen, Wei-Lun Liu, Yi-Ming Arthur Chen.

**Data curation:** I-An Jen, Wei-You Li, Yung-Feng Yen.

**Formal analysis:** I-An Jen, Wei-You Li, Patricia M. T. Chen, Yi-Ming Arthur Chen.

**Funding acquisition:** Yi-Ming Arthur Chen.

**Investigation:** Wei-Lun Liu, Yung-Feng Yen, Yi-Ming Arthur Chen.

**Methodology:** I-An Jen, Wei-You Li, Shang-Jung Wu, Cathy Shen Jang Fann.

**Project administration:** Yi-Ming Arthur Chen.

**Software:** Shang-Jung Wu, Cathy Shen Jang Fann.

**Supervision:** Yi-Ming Arthur Chen.

**Writing – original draft:** I-An Jen, Wei-You Li, Patricia M. T. Chen.

**Writing – review & editing:** Wei-You Li, Yi-Ming Arthur Chen.

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
