## [Decision Letter · Decision Letter 0]

4 Nov 2024

PONE-D-24-19695Molecular Epidemiology and 16-Year Survival Analysis of Cohort of HIV-1/AIDS Patients Infected with CRF07_BC in TaiwanPLOS ONE

Dear Dr. Chen,

Thank you for submitting your manuscript to PLOS ONE. After careful consideration, we feel that it has merit but does not fully meet PLOS ONE’s publication criteria as it currently stands. Therefore, we invite you to submit a revised version of the manuscript that addresses the points raised during the review process. We normally require two review reports to make a decision, but since Reviewer 1 has recommended a major revision, we might save time in the review process if you act on it now. Should you choose to revise and resubmit, we would invite Reviewer 1 and another reviewer to report on the revised manuscript. Please submit your revised manuscript by Dec 19 2024 11:59PM. If you will need more time than this to complete your revisions, please reply to this message or contact the journal office at plosone@plos.org . Please include the following items when submitting your revised manuscript:

We look forward to receiving your revised manuscript.

Kind regards,

Siew Ann Cheong, Ph.D.

Academic Editor

PLOS ONE

Journal Requirements:

3. Thank you for stating the following financial disclosure: “This work was sponsored by the National Science and Technology Council, Taiwan, R.O.C. under Grant no. MOST 111-2327-B-016 -001 and by the Fu Jen Catholic University, Taiwan, R.O.C.

under Grant no. 913M332-01.”

4. We note that you have indicated that there are restrictions to data sharing for this study. PLOS only allows data to be available upon request if there are legal or ethical restrictions on sharing data publicly. For more information on unacceptable data access restrictions, please see http://journals.plos.org/plosone/s/data-availability#loc-unacceptable-data-access-restrictions. Before we proceed with your manuscript, please address the following prompts: a) If there are ethical or legal restrictions on sharing a de-identified data set, please explain them in detail (e.g., data contain potentially identifying or sensitive patient information, data are owned by a third-party organization, etc.) and who has imposed them (e.g., a Research Ethics Committee or Institutional Review Board, etc.). Please also provide contact information for a data access committee, ethics committee, or other institutional body to which data requests may be sent. b) If there are no restrictions, please upload the minimal anonymized data set necessary to replicate your study findings to a stable, public repository and provide us with the relevant URLs, DOIs, or accession numbers. For a list of recommended repositories, please see https://journals.plos.org/plosone/s/recommended-repositories. You also have the option of uploading the data as Supporting Information files, but we would recommend depositing data directly to a data repository if possible. We will update your Data Availability statement on your behalf to reflect the information you provide.

Reviewers' comments:

Reviewer's Responses to Questions

**Comments to the Author**

1. Is the manuscript technically sound, and do the data support the conclusions?

Reviewer #1: Yes

2. Has the statistical analysis been performed appropriately and rigorously? 

Reviewer #1: No

3. Have the authors made all data underlying the findings in their manuscript fully available?

Reviewer #1: Yes

4. Is the manuscript presented in an intelligible fashion and written in standard English?

Reviewer #1: Yes

5. Review Comments to the Author

Reviewer #1: This is an important paper that looks at difference in clinical outcomes among different HIV subypes in Taiwan. The strength of the study is in the large number of patients it looks at that have prospective data. One major confounder is that CRF07_BC is more common among PWID while subtype B and CRF01_AE are more common in among those who acquired HIV through sexual contact. Nevertheless, the possibility of comparing subtype characteristic and clinical outcomes in the same geographic location is a rare opportunity to study non-B subtypes which are not well represented in literature. I do have a few concerns about the finding and the assumptions the authors have made:

1. The statement in the introduction "...since HIV-1 viral load correlates strongly with AIDS disease progression (lines 97 to 98)" is not supported by any citation, and is not consistent with published literature as shown for instance in Rodríguez et al. Predictive Value of Plasma HIV RNA Level on Rate of CD4 T-Cell Decline in Untreated HIV Infection. JAMA. 2006;296(12):1498–1506. doi:10.1001/jama.296.12.1498 (https://jamanetwork.com/journals/jama/fullarticle/203435). If there is newer evidence supporting the authors' assertion, this should be cited.

2. The methodology for determining subtypes is not nucleic acid sequence-based and uses an old method based on PCR product lengths. This can be problematic in an epidemics with multiple subtypes, as it is likely to misclassify recombinants between circulating subtypes depending on the recombination sites. Is it possible to sequence representative samples from the biobanks to verify that the subtyping is accurate?

3. Line 131 to 133 in the methodology states, "Once multiplex PCR showed two or more HIV-1 subtypes suggesting dual infection, serial PCR using a single pair of subtype-specific primers was used to confirm dual infection." I have looked through the results and I have not found any statement on whether dual infections were detected, and what significance these had on the findings. Moreover, finding two or more subtypes on the PCR products may be because of recombinants between circulating subtypes and not necessarily dual infection. Were steps made to differentiate this?

3. The higher death rate for CRF07_BC versus subtype B seems to be more of a function of the mode of transmission (PWID vs, MSM) as well as lower ART uptake than actual subtype differences. Was an attempt made to correct for these confounders through multivariate analysis and modelling to see if HIV subtype had a significant contribution to survival?

4. Lines 229 to 236, please generate hazard ratios with one of the subtypes as the reference and include 95% confidence intervals to see if the survival rates are significant or if there is overlap.

The low number of CRF01_AE samples means there will be a wide confidence interval and the seemingly low survival in non-ART treated patients may not be significant. Having said that, there is literature that suggests that CRF01_AE is associated with faster disease progression and higher viral loads so this may be a real subtype-related effect.

5. Lines 238 to 243, please show hazard ratios and 95% confidence intervals (and p values if preferred) instead of stating the differences were "statistically significant."

6. Lines 289-290 "Interesting (sic), if we don’t consider the risk factors of transmission routes first, we found that overall,

CRF07_BC has a significantly higher death rate (8.2% vs. 22.8%) compared to subtype B patients.

Please fix this line since the grammar is wrong (use Interestingly), and the death rates stated are confusing - is 8.2% for CRF07 or B? If you mean for CRF07, then it should be (22.8% vs. 8.2%) so as to make the order of statement of subtypes more consistent. Moreover, it is clear that more B than CRF07 were on ART so this should be considered as well. A proper multivariate analysis and specific univariate analyses will clarify the strengths of these associations.

7. Line 319 to 332, the conclusions are not tight and coherent. Please indicate the mean or (better) median viral loads and 95% confidence intervals for each subtype in the text when stating that CRF07 has a lower viral load than subtype B. If you look at Supplemental Table 1, the p value for comparisons of mean viral loads is not significant (p=0.359). Given the wide nature of viral loads, using median count, a log transformation or binary conditions (less than 100,000 copies versus >100,000 copies) may show more significant differences.

8. The conclusion of higher death rates (lines 289 to 290) for CRF07 is not consistent with the conclusion of slower disease progression due to lower viral loads (lines 320 to 322) and better survival. This needs to be resolved.

9. There are many grammatical and typographical errors throughout the manuscript. It is recommended that the revised manuscript be subjected to professional copyediting before resubmission.

6. PLOS authors have the option to publish the peer review history of their article (what does this mean? ). If published, this will include your full peer review and any attached files.

**Do you want your identity to be public for this peer review?** For information about this choice, including consent withdrawal, please see our Privacy Policy .

Reviewer #1: No

---

## [Author Response · Author response to Decision Letter 1]

6 Dec 2024

November 29, 2024

Dear Editor:

Thank you for your messages and the reviewer’s comments. We found the reviewer’s comments insightful and very helpful. We are pleased and encouraged by the reviewer’s general comments to our manuscript-“the strength of the study is in the large number of patients it looks at that have prospective data” and “the possibility of comparing subtype characteristic and clinical outcomes in the same geographic location is a rare opportunity to study non-B subtypes which are not well represented in literature”. As requested by the reviewer, we have further enhanced and double-checked the statistical analysis, with the help of two scholars who were added in the authorship. Attached please find our revised article entitled “Molecular Epidemiology and Long-term Survival Analysis of HIV-1/AIDS Patients Infected with CRF07_BC, CRF01_AE and Subtype B in Taiwan” and the following point-by-point reply to the reviewer #1.

Reviewer #1: This is an important paper that looks at difference in clinical outcomes among different HIV subtypes in Taiwan. The strength of the study is in the large number of patients it looks at that have prospective data. One major confounder is that CRF07_BC is more common among PWID while subtype B and CRF01_AE are more common in among those who acquired HIV through sexual contact. Nevertheless, the possibility of comparing subtype characteristic and clinical outcomes in the same geographic location is a rare opportunity to study non-B subtypes which are not well represented in literature. I do have a few concerns about the finding and the assumptions the authors have made:

1. The statement in the introduction "...since HIV-1 viral load correlates strongly with AIDS disease progression (lines 97 to 98)" is not supported by any citation, and is not consistent with published literature as shown for instance in Rodríguez et al. Predictive Value of Plasma HIV RNA Level on Rate of CD4 T-Cell Decline in Untreated HIV Infection. JAMA. 2006;296(12):1498–1506.doi:10.1001/jama.296.12.1498 (https://jamanetwork.com/journals/jama/fullarticle/203435). If there is newer evidence supporting the authors' assertion, this should be cited.

1. A: Thank you for your comments. We have modified the description in the introduction (Since the extent of viremia, measured as HIV-1 RNA viral load, has been postulated to be the best available surrogate marker of HIV-1 disease progression11, we hypothesized that patients infected with CRF07_BC might have better survival than patients infected with subtype B if they have similar risk of transmission (lines 96-100) and add a reference (No. 11: J W Mellors 1, C R Rinaldo Jr, P Gupta, R M White, J A Todd, L A Kingsley. Prognosis in HIV-1 infection predicted by the quantity of virus in plasma. Comment Science. 1996 May 24;272(5265):1167-70.).

2. The methodology for determining subtypes is not nucleic acid sequence-based and uses an old method based on PCR product lengths. This can be problematic in an epidemics with multiple subtypes, as it is likely to misclassify recombinants between circulating subtypes depending on the recombination sites. Is it possible to sequence representative samples from the biobanks to verify that the subtyping is accurate?

A: Thank you for your comments. We not only used nested multiplex PCR, but also employed nucleic acid sequencing with phylogenetic tree analysis of representative samples to determine the genotypes of our patients. We have clarified this issue and add two sentences in the materials and methods (lines 125-126; lines 133-134).

3-1. Line 131 to 133 in the methodology states, "Once multiplex PCR showed two or more HIV-1 subtypes suggesting dual infection, serial PCR using a single pair of subtype-specific primers was used to confirm dual infection." I have looked through the results and I have not found any statement on whether dual infections were detected, and what significance these had on the findings. Moreover, finding two or more subtypes on the PCR products may be because of recombinants between circulating subtypes and not necessarily dual infection. Were steps made to differentiate this?

A: Thank you for your enquiry. Yes, we did find two patients who were MSM have dual infection with Subtype B and CRF01_AE. These two patients were excluded from our survival analysis to avoid confusion. Since we did not include these two patients for survival analysis, we did not conduct long-distance PCR combined with cloning and sequencing to differentiate the situation.

3-2. The higher death rate for CRF07_BC versus subtype B seems to be more of a function of the mode of transmission (PWID vs, MSM) as well as lower ART uptake than actual subtype differences. Was an attempt made to correct for these confounders through multivariate analysis and modelling to see if HIV subtype had a significant contribution to survival?

A: We fully agreed with your comments. We took another approach to correct these confounders through comparing the survival between patients infected with different subtypes in the same risk group. As shown in Figure 2F, we found that among IDUs, patients infected with CRF07_BC had better survival rate (74.3%) than patients infected with subtype B (45.7%) and the difference was statistically significant. These findings suggest that the transmission route is one major factor influencing death rate. By eliminating this confounder and comparing only patients with the same transmission route, we can more accurately assess the true impact of subtypes on survival outcomes. We have added these sentences in the discussion of the manuscript (lines 363-368).

4. Lines 229 to 236, please generate hazard ratios with one of the subtypes as the reference and include 95% confidence intervals to see if the survival rates are significant or if there is overlap.

The low number of CRF01_AE samples means there will be a wide confidence interval and the seemingly low survival in non-ART treated patients may not be significant. Having said that, there is literature that suggests that CRF01_AE is associated with faster disease progression and higher viral loads so this may be a real subtype-related effect.

A: Thanks for the comments. We have generated hazard ratios with 95% confidence intervals and added a new Table (Table 7). The results show the hazard ratio for the 16-year death rate being 3.84 (95% Confidence Interval: 2.68-5.49) for CRF07_BC and 5.43 (95% Confidence Interval: 2.78-10.64) for CRF01_AE, compared with subtype B. (lines 247-248).

5. Lines 238 to 243, please show hazard ratios and 95% confidence intervals (and p values if preferred) instead of stating the differences were "statistically significant."

A: Our permission of using the database from the National Health Insurance Database had expired. Therefore, we no longer can access the raw data for the analysis suggested by the reviewer. However, the log-rank test results show that the death rates among the three main subtypes of HIV-1 in the intravenous drug user population in Taiwan are significantly different.

6. Lines 289-290 "Interesting (sic), if we don’t consider the risk factors of transmission routes first, we found that overall,

CRF07_BC has a significantly higher death rate (8.2% vs. 22.8%) compared to subtype B patients.

Please fix this line since the grammar is wrong (use Interestingly), and the death rates stated are confusing - is 8.2% for CRF07 or B? If you mean for CRF07, then it should be (22.8% vs. 8.2%) so as to make the order of statement of subtypes more consistent. Moreover, it is clear that more B than CRF07 were on ART so this should be considered as well. A proper multivariate analysis and specific univariate analyses will clarify the strengths of these associations.

A: Thanks for your suggestion, we have corrected our statement in the manuscript (lines 294-295). Unfortunately, due to the reason mentioned above, we are no longer able to access the raw data to conduct the statistical analysis.

7. Line 319 to 332, the conclusions are not tight and coherent. Please indicate the mean or (better) median viral loads and 95% confidence intervals for each subtype in the text when stating that CRF07 has a lower viral load than subtype B. If you look at Supplemental Table 1, the p value for comparisons of mean viral loads is not significant (p=0.359). Given the wide nature of viral loads, using median count, a log transformation or binary conditions (less than 100,000 copies versus >100,000 copies) may show more significant differences.

A: Thanks for your suggestion. Based on our current data and your suggestions, we have modified the sentences in the discussion (lines 324 to 325).

8. The conclusion of higher death rates (lines 289 to 290) for CRF07 is not consistent with the conclusion of slower disease progression due to lower viral loads (lines 320 to 322) and better survival. This needs to be resolved.

A: Thank you for your comments. We have revised several sentences to make the conclusion clearer (lines 363-368).

9. There are many grammatical and typographical errors throughout the manuscript. It is recommended that the revised manuscript be subjected to professional copyediting before resubmission.

A. Thanks for your suggestions. We have hired professional editor to correct our manuscript.

---

## [Decision Letter · Decision Letter 1]

27 Feb 2025

PONE-D-24-19695R1Molecular Epidemiology and Long-term Survival Analysis of HIV-1/AIDS Patients Infected with CRF07_BC, CRF01_AE and Subtype B in TaiwanPLOS ONE

Dear Dr. Chen,

Thank you for submitting your manuscript to PLOS ONE. After careful consideration, we feel that it has merit but does not fully meet PLOS ONE’s publication criteria as it currently stands. Therefore, we invite you to submit a revised version of the manuscript that addresses the points raised during the review process.

In particular, please address the concerns of Reviewer 2 on the statistical analysis done in the manuscript, and the other minor comments.

We look forward to receiving your revised manuscript.

Kind regards,

Siew Ann Cheong, Ph.D.

Academic Editor

PLOS ONE

Reviewers' comments:

Reviewer's Responses to Questions

**Comments to the Author**

1. If the authors have adequately addressed your comments raised in a previous round of review and you feel that this manuscript is now acceptable for publication, you may indicate that here to bypass the “Comments to the Author” section, enter your conflict of interest statement in the “Confidential to Editor” section, and submit your "Accept" recommendation.

Reviewer #1: All comments have been addressed

Reviewer #2: (No Response)

2. Is the manuscript technically sound, and do the data support the conclusions?

Reviewer #1: Yes

Reviewer #2: No

3. Has the statistical analysis been performed appropriately and rigorously? 

Reviewer #1: Yes

Reviewer #2: No

4. Have the authors made all data underlying the findings in their manuscript fully available?

Reviewer #1: No

Reviewer #2: Yes

5. Is the manuscript presented in an intelligible fashion and written in standard English?

Reviewer #1: Yes

Reviewer #2: Yes

6. Review Comments to the Author

Reviewer #1: The authors have addressed all my major concerns and their revisions are satisfactory. They have indicated that there are restrictions on accessing their data but have explained that they themselves do not own the data and were only given permission to access it for a limited time. I believe this is reasonable.

Reviewer #2: The paper presented by Jen et al. presents a unique opportunity to look at differences in clinical outcomes among HIV-1 epidemics in the one distinct geographical location. Particularly noteworthy is the inclusion and number of samples from a variety of non-subtype B HIV-1 samples; the authors should be commended for this focus.

However, I have major concerns regarding the statistics and analysis performed on these samples, primarily in regard to drawing any conclusions based on long-term survival without considering and/or stratifying for antiretroviral therapy (ART) usage and adherence. ART is known to significantly increase survival rates. However, the authors also find that ART uptake is different among different epidemics (B vs. CRF07_BC). Therefore, this represents a major confounder of the results that significantly affects how the results are being presented and interpreted.

As such, I would suggest the following major revisions:

- The populations observed in the two epidemics (B and CRF07_BC) are very different, as noted in lines 178-185. Each epidemic has been found to have variations in subtype, route of transmission and ART uptake, and likely has different underlying nuances based on differences in demographics. I would use a mixed effect model or the like for this analysis, as the viral subtype is not independent of other characteristics that may impact long term survival. While I believe there has been some attempt to do this, particularly in figure 2, this approach to the data needs to be carried on throughout the entire analysis to ensure that the conclusions drawn are valid.

- Given the association with CRF07_BC with people who inject drugs in this epidemic, is there also an association in not just uptake, but also adherence to ART (i.e. achieving adequate viral suppression) for this subtype? This may also lead to differences in long-term survival. I do understand from the discussion (lines 357 – 363) that looking granularly at this may not be possible, however it does remain a significant issue that should be discussed.

- Lines 200-211: individuals from the subtype B epidemic were found to have a better treatment response to ART through monitoring CD4 levels over time. However, was this analysis limited to only individuals adhering to therapy? Again, while I appreciate this may not be possible, I believe this needs to be made more explicit as adherence may be a significant factor here, particularly among people who inject drugs.

- Table 4 and lines 214-226, the data does not seem to be stratified for ART usage. This is a major confounder of the results, given that ART significantly increases lifespan, something the authors also find in lines 241-242. This may be influencing the results presented here, as those individuals with a lower death rate also have the highest uptake of ART, though this is not taken into account.

- Lines 229-237, again, the data does not seem to be stratified for ART usage, even though a significant association between the different subtypes and ART uptake has been shown prior (lines 188-192).

I also have the following suggested minor revisions:

- Lines 69-70: ART is the key to controlling the epidemic; it is not a question of understanding (“being baffled”), but rather an issue of access to therapy.

- Lines 132-141: it is not immediately clear when the genotyping was performed, though I’m assuming from Figure 1 a blood draw was taken at/near the time of diagnosis. It may be helpful to make this explicit. While also unusual to not perform genotyping analysis by sequencing all samples (rather only a subset were sequenced), I do appreciate this is a historical data.

- Lines 205-206: long term non-progressors generally refers to individuals who are not taking ART, however I believe this paragraph/Table 3 only refers to individuals who are taking therapy.

Finally, though I appreciate it can be difficult, I would also encourage the authors to consider using person-centred language throughout the manuscript (e.g. people who inject drugs or people living with HIV) as per the UNAIDS terminology guidelines ( https://www.unaids.org/en/resources/documents/2024/terminology_guidelines)

7. PLOS authors have the option to publish the peer review history of their article (what does this mean? ). If published, this will include your full peer review and any attached files.

**Do you want your identity to be public for this peer review?** For information about this choice, including consent withdrawal, please see our Privacy Policy .

Reviewer #1: **Yes: ** Edsel Maurice T. Salvana

Reviewer #2: No

---

## [Author Response · Author response to Decision Letter 2]

25 Mar 2025

March 6, 2025

Dear Editor:

Thank you for your message and the reviewer’s comments. We appreciate reviewer #1’s favorable review and recommendation for publication. We also found the reviewer #2’s meticulous comments insightful and helpful. As requested by the reviewer, we have further revised our manuscript accordingly. Attached please find our revised article entitled “Molecular Epidemiology and Long-term Survival Analysis of HIV-1 and AIDS Patients Infected with CRF07_BC, CRF01_AE and Subtype B in Taiwan” and the following point-by-point reply to the reviewer #2.

Reviewer #2: The paper presented by Jen et al. presents a unique opportunity to look at differences in clinical outcomes among HIV-1 epidemics in the one distinct geographical location. Particularly noteworthy is the inclusion and number of samples from a variety of non-subtype B HIV-1 samples; the authors should be commended for this focus.

However, I have major concerns regarding the statistics and analysis performed on these samples, primarily in regard to drawing any conclusions based on long-term survival without considering and/or stratifying for antiretroviral therapy (ART) usage and adherence. ……

As such, I would suggest the following major revisions:

1. The populations observed in the two epidemics (B and CRF07_BC) are very different, as noted in lines 178-185. Each epidemic has been found to have variations in subtype, route of transmission and ART uptake, and likely has different underlying nuances based on differences in demographics. I would use a mixed effect model or the like for this analysis, as the viral subtype is not independent of other characteristics that may impact long term survival. While I believe there has been some attempt to do this, particularly in figure 2, this approach to the data needs to be carried on throughout the entire analysis to ensure that the conclusions drawn are valid.

Answer: We appreciate the reviewer’s thoughtful comment and agree that a mixed-effects model would be a valuable approach to analyze clustered data like ours. However, we regret to inform the reviewer that we no longer have access to the raw data required for such an analysis. Our permission to use the National Health Insurance Database has unfortunately expired, and as a result, we are unable to perform the suggested mixed-effects model analysis currently. We recognize the importance of this methodology and would certainly consider it in future studies, should access to the data be renewed. However, the conclusion we made in lines 335-338 (as presented below) suggests that by focusing on a specific risk group, with all subjects under similar environmental conditions—such as lifestyle and socioeconomic status—the true effect of viral subtypes on survival rates could be investigated more independently.

“Moreover, among the PWID who had been genotyped, CRF07_BC patients had better survival rate than patients infected with subtype B or CRF01_AE and the differences are statistically significant (Fig. 2F). These findings support our hypothesis that under the premise of same risk factor, a lower viral load in CRF07_BC patients can contribute to their slower disease progression.”

2. Given the association with CRF07_BC with people who inject drugs in this epidemic, is there also an association in not just uptake, but also adherence to ART (i.e. achieving adequate viral suppression) for this subtype? This may also lead to differences in long-term survival. I do understand from the discussion (lines 357 – 363) that looking granularly at this may not be possible, however it does remain a significant issue that should be discussed.

Lines 200-211: individuals from the subtype B epidemic were found to have a better treatment response to ART through monitoring CD4 levels over time. However, was this analysis limited to only individuals adhering to therapy? Again, while I appreciate this may not be possible, I believe this needs to be made more explicit as adherence may be a significant factor here, particularly among people who inject drugs.

Table 4 and lines 214-226, the data does not seem to be stratified for ART usage. This is a major confounder of the results, given that ART significantly increases lifespan, something the authors also find in lines 241-242. This may be influencing the results presented here, as those individuals with a lower death rate also have the highest uptake of ART, though this is not taken into account.

Lines 229-237, again, the data does not seem to be stratified for ART usage, even though a significant association between the different subtypes and ART uptake has been shown prior (lines 188-192).

Answer: We appreciate the reviewer’s comment. We agree that adherence to ART is an important factor for long-term survival. Unfortunately, again, we no longer have access to the raw data required for such an analysis. Our permission to use the National Health Insurance Database has unfortunately expired, and as a result, we are unable to perform the analyses suggested by the reviewer. However, to clearly notify readers of this major point, we added “Assuming these individuals stringently adhere to their prescribed ART,” (Line 202-03) in the Results section, and “Lastly, given the adherence to ART is a known determinant for HIV-1 treatment outcome, particularly in IDUs, such information were unfortunately unavailable for our patients.”(Lines 365-67) in the Discussion section.

3. I also have the following suggested minor revisions:

3-1. Lines 69-70: ART is the key to controlling the epidemic; it is not a question of understanding (“being baffled”), but rather an issue of access to therapy.

Answer: We appreciate the reviewer’s comment. We have changed our sentence to “disease control of human immunodeficiency virus type 1 (HIV-1) remains manifestly inadequate.” (Line 70)

3-2 Lines 132-141: it is not immediately clear when the genotyping was performed, though I’m assuming from Figure 1 a blood draw was taken at/near the time of diagnosis. It may be helpful to make this explicit. While also unusual to not perform genotyping analysis by sequencing all samples (rather only a subset were sequenced), I do appreciate this is a historical data.

Answer: We appreciate the reviewer’s comment. To clarify the timing of blood draw and genotyping, we have modified our sentence to “blood samples from participants were collected (at the time of diagnosis).” (Line 134)

- Lines 205-206: long term non-progressors generally refers to individuals who are not taking ART, however I believe this paragraph/Table 3 only refers to individuals who are taking therapy.

Answer: We appreciate the reviewer’s comment. To clarify, we have modified our sentence to “for subtype B patients who received ART increased 13.8% (from 55.5% to 69.3%).” (Line 208)

4. Finally, though I appreciate it can be difficult, I would also encourage the authors to consider using person-centered language throughout the manuscript (e.g. people who inject drugs or people living with HIV) as per the UNAIDS terminology guidelines ( https://www.unaids.org/en/resources/documents/2024/terminology_guidelines)

Answer: We appreciate the reviewer’s comment. We agree the importance of using person-centered language. We have adapted person-centered language and modified the entire manuscript as per the UNAIDS terminology guidelines, which was cited in the acknowledgement. (Line 387-88)

---

## [Editor Report · Decision Letter 2]

7 Apr 2025

Molecular Epidemiology and Long-term Survival Analysis of HIV-1/AIDS Patients Infected with CRF07_BC, CRF01_AE and Subtype B in Taiwan

PONE-D-24-19695R2

Dear Dr. Chen,

We’re pleased to inform you that your manuscript has been judged scientifically suitable for publication and will be formally accepted for publication once it meets all outstanding technical requirements.

Kind regards,

Siew Ann Cheong, Ph.D.

Academic Editor

PLOS ONE
---

## [Editor Report · Acceptance letter]

PONE-D-24-19695R2

PLOS ONE

Dear Dr. Chen,

I'm pleased to inform you that your manuscript has been deemed suitable for publication in PLOS ONE. Congratulations! Your manuscript is now being handed over to our production team.

Kind regards,

on behalf of

Dr. Siew Ann Cheong

Academic Editor

PLOS ONE